# Optimization Design and Experimental Verification for the Mixed-Flow Fan of a Stratospheric Airship

Wei Qu , Wentao Gong, Chen Chen, Taihua Zhang * and Zeqing He

Aerospace Information Research Institute, Chinese Academy of Sciences, Beijing 100094, China
* Correspondence: zth@aoe.ac.cn

**Abstract:** Large-flow, high-pressure fans are required to improve the shape-keeping and flight-height-adjusting ability of stratospheric airships. This study optimizes and analyzes a fan MIX-130 suitable for a stratospheric airship. Five design parameters are selected to optimize the fan's static pressure rise and efficiency: impeller outlet installation angle, installation angle increment, blade thickness, diffuser tilt angle, and $L_{16}(4^5)$ orthogonal test for optimization research. Based on the optimization results, the fan is processed, a fan test bench is built to verify the accuracy of the fan numerical analysis method, and the fan's performance curve in the stratosphere environment is given. The results demonstrate that after optimization, the static pressure rise in the MIX-130 fan increases by 47.5%, and the efficiency increases by 8%. The performance test data of the MIX-130 fan are consistent with the numerical analysis results. Furthermore, the flow pressure curve is significantly improved compared with the existing fan, satisfying the requirements of airship flight missions. The fan structure optimization and testing methods presented in this manuscript can provide a reference for designing and testing stratospheric airship fans.

**Keywords:** near-space; stratospheric airship; orthogonal optimization; fan test; similarity principle



## 1. Introduction

A stratospheric airship, an aerostat that can stay in near space (18–22 km) for an extended duration, can carry several loads to achieve communication, navigation, observation, early warning, and other functions [1–3]. It has a high cost-effectiveness ratio, broad coverage, and strong load capacity, demonstrating broad application potential [4–6].

It is a unique control feature of stratospheric airships, overpressure balloons, and other types of aerostats to use the main and auxiliary airbags for height adjustment [7]. The fan is used to compress the air to the auxiliary airbag, which can reduce the airship's standing altitude. The valve is used to exhaust the air in the auxiliary airbag, and the airship's standing altitude can be raised [8,9]. By changing the airship height, the wind layer at different altitudes is used to adjust the flight path, achieve specific trajectory control objectives, and complete the designated flight mission [10,11].

Mature aviation fans are used on stratospheric airships, such as MAX45016 and MAX80001 from AMETEK [12,13]. With the continuous development and progress of stratospheric airship technology, the demands of cross-day and night flight, wide-range cruise, fixed point hover, and other tasks are increasing, and the performance of the existing fans cannot meet the task requirements for three reasons. First, at the stratospheric airship standing altitude, the pressure head of the aviation fan is low, which cannot overcome the airship's internal pressure to fill with air [14,15]; thus, the airship cannot adjust its height through the auxiliary airbag during the day. Second, at the stratospheric airship standing altitude, the flow rate of the aviation fan is low. The pressure will decrease sharply if the airship encounters cold clouds [16]. The flow rate of the aviation fan is insufficient to quickly fill the air to maintain the aerodynamic shape of the stratospheric airship. Third, the fans of stratospheric airships are usually used with valves. Because of the limitation of

the shape of aviation fans, the valves can only be installed at the outlet of the fans, which will significantly reduce fan efficiency [17].

Compared with traditional fans, stratospheric fans have several design difficulties.

(a) Harsh operating conditions: severe changes in atmospheric density, pressure, and temperature from the ground to the stratosphere, the large temperature difference between day and night at the standing altitude of the airship, low Reynolds number, and large drag coefficient [18].

(b) Power consumption limitations: The energy of stratospheric airships comes from onboard lithium batteries and solar arrays [19], and the energy supply is limited, so the power consumption of fans is limited.

(c) Weight constraints: the stratospheric airship platform itself requires a lightweight design, so the fan must be as lightweight as possible.

(d) High-performance requirements: in the stratospheric, low-temperature, low-pressure environment, the fan must have sufficient pressure head and flow rate but also high efficiency to save energy consumption. Satisfying these requirements necessitates relevant design and research for a stratospheric airship fan.

Sun et al. [20] designed centrifugal and axial fans for stratospheric conditions. The variation rule of fan characteristics of airship during descent is compared. Wei et al. [21] proposed a selection method for a centrifugal fan for an aerostat, which can provide a reference for the selection of aerostat fans. Zhao et al. [22] studied the operational characteristics of centrifugal fans at different altitudes by considering the changes in air density, pressure, and temperature in high-altitude environments. Yang et al. [23] studied the flow field characteristics of a high-flow axial fan in stratospheric environments using computational fluid dynamics (CFD). The results demonstrated that selecting a guide vane and diffuser would significantly affect fan efficiency. Zhang et al. [24] studied the relationship between the altitude, the pressure difference of the aerostat capsule, the number of fans used, and the fan inflation efficiency. The results demonstrated that it was feasible to use multiple fans in series or parallel for different aerostat flight tasks. In general, there is a lack of research in the field of stratospheric airship fans.

The core component of the fan is the impeller, which is related to the performance of the entire fan. It is necessary to optimize the impeller design to produce an efficient fan. Scholars have performed extensive research in this field. Fan et al. [25] obtained the optimal combination of impellers through optimization design, combined with range analysis and variance analysis, and significantly improved the aerodynamic performance of the fan. Wang et al. [26] used the number of blades and the blade outlet angle as variables to optimize the fan impeller parameters through the least squares method and obtained promising results. Xu et al. [27] adopted the orthogonal optimization method to optimize the design of a centrifugal pump, significantly improving its efficiency and head. Wang et al. [28] adopted an artificial intelligence optimization algorithm to carry out a multi-objective optimization design of a centrifugal pump, and the performance of the centrifugal pump has been significantly improved. Jung et al. [29] studied the influence of structural parameters of a mixed-flow fan on fan performance and analyzed the influence of a single parameter on performance using CFD. The research found that the diffuser structure would significantly affect fan performance. Wang et al. [30] adopted the combinatorial optimization system to improve the performance of mixed-flow pumps and achieved promising results.

Previous studies on stratospheric fans and an optimization method for fan impellers are relevant but have limitations, focused primarily on the theoretical derivation and simulation analysis. The weight and power consumption of the fan involved do not apply to stratospheric airships. Furthermore, the main body of research in most papers includes components that are not consistent with the use of the stratospheric airship environment, such as a water pump or industrial fan.

In this study, a stratospheric mixed-flow fan [31] is used as the research object, and the improvement of fan efficiency and static pressure rise is used as the optimization objectives.

The $L_{16}(4^5)$ orthogonal test is adopted to conduct the multi-parameter and multi-objective optimization study on the fan. Finally, the accuracy of the numerical analysis method and the rationality of the fan design are verified by combining the performance tests. The fan structure optimization and testing methods presented in this manuscript can provide a reference for designing and testing stratospheric airship fans.

The rest of the manuscript is organized as follows: Section 2 describes the research object of the MIX-130 fan and the numerical analysis method and boundary conditions adopted in this manuscript. In Section 3, the optimal design of the MIX-130 fan is carried out by the orthogonal optimization method, and the optimization results are analyzed. In Section 4, the performances of the MIX-130 fan are carried out to verify the accuracy of the numerical analysis method and the rationality of the fan design. Section 5 concludes the manuscript and provides future perspectives.

## 2. Numerical Methods and Boundary Conditions

### 2.1. Numerical Method

Traditional aviation fans, such as AMETEK's MAX45016 and MAX80001 [8,9], adopt an axial impeller design. Because of the saddle shape area on the performance curve of the axial fan, the available effective range is small and is unsuitable for the high-flow-rate, high-pressure head of stratospheric airships.

The mixed-flow impeller can consider the characteristics of the high-pressure head of a centrifugal fan and the large flow of an axial flow fan [32]. The vaneless diffuser has a wide flow range and simple structure [33], so the fan structure design with the mixed impeller and vaneless diffuser is more appropriate for the variable working conditions of the stratospheric airship fan. Figure 1 illustrates the MIX-130 fan, the object of this study.

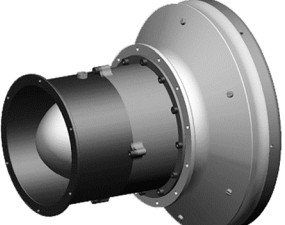

**Figure 1.** MIX-130 fan.

The initial design parameters of the MIX-130 fan are listed in Table 1.

**Table 1.** Initial design parameters of the MIX-130 fan.

| Description | Parameter | Value |
|---|---|---|
| Design flow rate | Q | 1100 $m^3 \cdot h^{-1}$ |
| Static pressure rise | $P_R$ | 700 Pa |
| Rated speed | n | 25,000 Rpm |
| Power output | P | $\not> $500 W |
| Blade number of impeller | Bn | 18 |
| Blade tip clearance | / | 0.5 mm |
| Blade thickness | t | 1.5 mm |
| Diameter of impeller in | $D_2$ | 80 mm |
| Diameter of impeller out | $B_2$ | 130 mm |
| Blade angle at leading edge | $\beta_1$ | 39–30° |
| Blade angle at trailing edge | β | 45–49° |
| Diffuser tilt angle | φ | 57° |
| Voltage | V | 16–32 V |
| Total weight | / | $\not> $2.5 kg |

Based on the structure of the MIX-130 fan, its fluid domain can be extracted, as depicted in Figure 2. The fluid domain includes three parts: the inlet pipe, impeller, and diffuser.

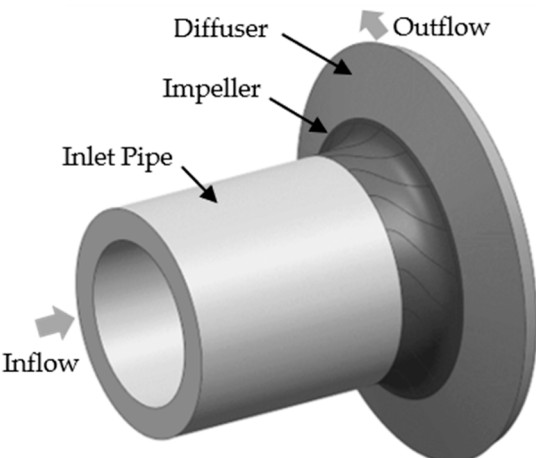

**Figure 2.** The fluid domain of MIX-130 fan.

In this study, fluid dynamics analysis software ANSYS CFX is used to analyze the fluid characteristics of the fan. Because the flow inside the fan is predominantly complex turbulent flow, an appropriate turbulence solution method must be selected to accurately simulate the flow of the air inside the fan [34]. Based on existing studies, the shear stress transport model (SST) [18,35] is more appropriate for analyzing a high altitude and a low Reynolds number. Therefore, the SST model is selected to solve the three-dimensional time-average Navier-Stokes equation.

Given the symmetrical characteristics of the fluid domain of the fan, the cyclic symmetric boundary is established, and a 1/18 flow passage is adopted for the numerical analysis to save calculation time, as depicted in Figure 3.

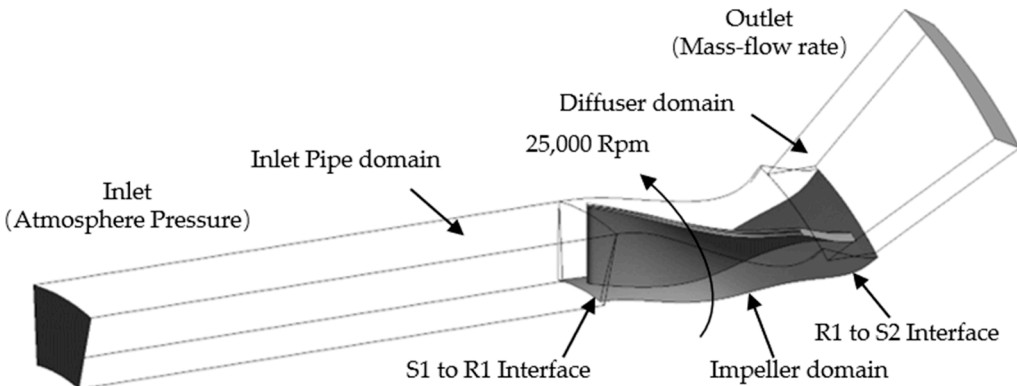

**Figure 3.** Computational domain.

The impeller is set as a rotating area with a speed of 25,000 Rpm, and the inlet tube and diffuser part are set as a stationary area. "Mixing-plane" technology is adopted to connect the interface between the rotary static interface, which axially averages the parameters of the interface of the upper-level components and then transmits them to the interface of the lower-level components [36]. All walls are in a slip-free condition. The calculation is considered convergent in steady-state computation when the root mean square residual is less than $10^{-5}$.

### 2.2. Boundary Conditions

The boundary conditions of the MIX-130 fan simulation analysis should be consistent with the real-world environment. The specific value is related to the altitude of the airship

target. The atmospheric temperature, pressure, and density in the high air can be calculated according to the ISA (International Standard Atmosphere) model [9] as follows:

$$
T = \begin{cases}
288.15 - 0.0065h & (0 \leq h \leq 11000) \\
216.65 & (11000 < h \leq 21000) \\
288.15(0.682457 + h/288153.5) & (21000 < h \leq 32000)
\end{cases} \tag{1}
$$

$$
P = \begin{cases}
101325(1 - 0.0065h/288.15)^{5.25588} & (0 \leq h \leq 11000) \\
22631.8e^{1.73-0.000157 \times h} & (11000 < h \leq 21000) \\
101325(0.988626 + h/198915)^{-34.16319} & (21000 < h \leq 32000)
\end{cases} \tag{2}
$$

$$
\rho = \begin{cases}
1.22505(1 - 0.0065h/288.15)^{4.25588} & (0 \leq h \leq 11000) \\
0.36392e^{1.73-0.000157 \times h} & (11000 < h \leq 21000) \\
1.22505 (0.988626 + h/201161)^{-35.16319} & (21000 < h \leq 32000)
\end{cases} \tag{3}
$$

As defined by Equations (1)–(3), the environmental parameters corresponding to the airship's standing altitude are as follows: atmospheric temperature of 216.65 K, the atmospheric pressure of 5500 Pa, and atmospheric density of 0.094 kg.m$^{-3}$.

The numerical analysis boundary conditions of the fan are presented in Table 2. The total pressure boundary pertains to the fan inlet, and the mass flow rate boundary pertains to the fan outlet.

**Table 2.** Numerical analysis of boundary conditions for MIX-130 fan.

| Boundary | Setting | Value |
|---|---|---|
| S1 Inlet | Total Pressure | 5500 Pa |
| S1 Inlet | Total Temperature | 216.65 K |
| S2 Outlet | Mass Flow Rate | 0.0288 Kg.s$^{-1}$ |
| R1 Speed | / | 25,000 Rpm |
| Reference Pressure | / | 0 Pa |
| Residual | RMS | $1 \times 10^{-5}$ |

### 2.3. Meshing and Independence

The fan inlet pipe and diffuser are meshed by ANSYS Mechanic. The mesh structure of the impeller is generated by ANSYS Turbo Mesh. As depicted in Figure 4, all the meshes are hexahedral.

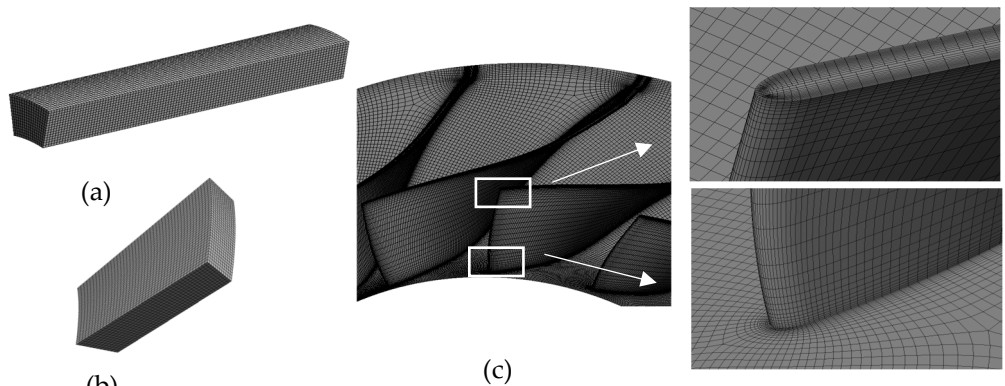

(a)

(b)

(c)

**Figure 4.** Computational mesh: (**a**) inlet pipe; (**b**) diffuser; (**c**) impeller.

The number of meshes will directly affect the accuracy of simulation results. Choosing a reasonable number of meshes can save calculation time, so it is critical to verify the mesh independence. In this study, six groups of different mesh numbers are selected for the mesh independence test. Table 3 presents the mesh distribution and independence test results.

**Table 3.** Mesh independence test.

| Item | Mesh1 | Mesh2 | Mesh3 | Mesh4 | Mesh5 | Mesh6 |
|------|-------|-------|-------|-------|-------|-------|
| Inlet pipe | 3520 | 16,744 | 26,400 | 26,400 | 26,400 | 10,080 |
| Impeller | 105,100 | 256,300 | 315,350 | 393,050 | 522,150 | 630,300 |
| Diffuser | 3059 | 19,008 | 10,080 | 10,080 | 10,080 | 10,080 |
| Total mesh | 111,679 | 292,052 | 351,830 | 429,530 | 558,630 | 650,460 |
| PRi/PR1 | 1 | 1.015 | 1.016 | 1.017 | 1.019 | 1.018 |
| ηi/η1 | 1 | 1.017 | 1.017 | 1.017 | 1.019 | 1.018 |

Based on the analysis results, when the inlet pipe mesh number is 26,400, the impeller mesh number is 522,150, the diffuser mesh number is 10,080, and the fan efficiency and static pressure ratio change rates are small. Therefore, in the subsequent calculation and analysis, it is reasonable to control the total mesh number at 558,630.

*2.4. Original Fan Performance*

Based on this analysis method, the efficiency of the MIX-130 fan is only 64%. The static pressure rise is only 511 Pa at the design condition point, which differs from the target design value and cannot meet the demand. Therefore, it is necessary to optimize the design and improve the fan structure.

**3. Optimal Fan Design**

*3.1. Optimization Parameters*

The impeller is the core energy conversion component of the fan. Its structural parameters directly affect the fan's pressure ratio and efficiency, so the optimization of the fan focuses primarily on the impeller.

The blade is the central part of the impeller. In terms of fluid, the thickness and number of blades directly determine the volume of the flow path in the impeller. If the number of blades is too small, secondary flow loss will occur in the impeller, reducing fan efficiency. If the number of blades is too large, the flow path of the impeller will be reduced, the flow loss and friction loss inside the impeller will be increased, and fan efficiency will be reduced [37]. In terms of structure, the thickness of the blade is too thick to increase the weight of the impeller, increasing the fan's power consumption. If the blade thickness is too thin, it will affect the strength of the impeller and increase the difficulty of processing. Therefore, blade thickness and blade number are important optimization parameters of the impeller. The blade outlet installation and variation angles directly determine the blade shape and significantly impact fan outlet pressure and efficiency [38]. Furthermore, the tilt angle of the diffuser in the MIX-130 fan is a special parameter. Its inclination angle will directly affect the structural size of the entire fan, so it is also selected as an optimization parameter—defined as the angle between the center line of the diffuser and the axial direction of the fan.

In this study, five parameters, including the number of blades, Bn, thickness, t, outlet installation angle, β, outlet installation angle increment, Δβ, and the diffuser tilt angle, φ, are selected for optimization. In addition to the above parameters, other impeller parameters, such as impeller inlet and outlet diameter, width, and blade tip clearance [39], are limited by the overall fan structure design and remain the same as the original impeller without change.

*3.2. Orthogonal Test Analysis*

The orthogonal test method is a scientific method to study the multi-factor optimization test by applying the orthogonal principle of an orthogonal table and mathematical statistics analysis. It can optimize the combination of optimal parameters or conditions of each factor with the least number of tests [25] and is highly effective for optimizing the impeller design [38]. The orthogonal table consists of two parts: factor and factor level. The

test individuals in the orthogonal table are symmetrical and evenly distributed to ensure that the results are representative of all factor combinations.

Factor levels were determined concerning the original impeller and design experience. In the five design factors selected in this study, each factor changes at four levels. For improving the static pressure rise and efficiency of the fan, the $L_{16}(4^5)$ orthogonal test was used to optimize the fan design. The factors and levels of the orthogonal test are presented in Table 4. A blade thickness *t* of 1~2 indicates that the blade thickness is 2 mm at the Hub and 1 mm at the shroud, and the thickness of the middle part changes linearly.

**Table 4.** Orthogonal test factors and levels.

| Level/Factor | Bn | β/° | Δβ/° | t/mm | ϕ/° |
|---|---|---|---|---|---|
| 1 | 17 | 35 | 1 | 1 | 57 |
| 2 | 18 | 40 | 2 | 1.5 | 60 |
| 3 | 19 | 45 | 3 | 2 | 63 |
| 4 | 20 | 50 | 4 | 1~2 | 68 |

As presented in Table 4, orthogonal Table 5 can be obtained with 16 groups of fan models with different factors and levels. The numerical method in Section 2 is used to analyze each group of fan models.

**Table 5.** Orthogonal table.

| Level/Factor | Bn | β/° | Δβ/° | t/mm | ϕ/° |
|---|---|---|---|---|---|
| 1 | 17 | 35 | 1 | 1 | 57 |
| 2 | 17 | 40 | 2 | 1.5 | 60 |
| 3 | 17 | 45 | 3 | 2 | 63 |
| 4 | 17 | 50 | 4 | 1~2 | 68 |
| 5 | 18 | 35 | 2 | 1~2 | 63 |
| 6 | 18 | 40 | 1 | 2 | 68 |
| 7 | 18 | 45 | 4 | 1.5 | 57 |
| 8 | 18 | 50 | 3 | 1 | 60 |
| 9 | 19 | 35 | 3 | 1.5 | 68 |
| 10 | 19 | 40 | 4 | 1 | 63 |
| 11 | 19 | 45 | 1 | 1~2 | 60 |
| 12 | 19 | 50 | 2 | 2 | 57 |
| 13 | 20 | 35 | 4 | 2 | 60 |
| 14 | 20 | 40 | 3 | 1~2 | 57 |
| 15 | 20 | 45 | 2 | 1 | 68 |
| 16 | 20 | 50 | 1 | 1.5 | 63 |

*3.3. Optimization Results and Discussion*

The static pressure rise and efficiency of 16 groups of different fans at the design point are presented in Table 6.

The static pressure rise in the fan is defined as:

$$P_R = P_{outlet} - P_{inlet} \tag{4}$$

$P_R$ is the static pressure rise, $P_{outlet}$ is the static pressure at the fan outlet, and $P_{inlet}$ is the static pressure at the fan inlet.

The fan efficiency is defined as:

$$\eta = P_R \cdot Q / T \cdot W \tag{5}$$

$P_R$ is the static pressure rise, $Q$ is the flow rate, $T$ is the impeller torque, and $W$ is the impeller speed.

**Table 6.** Static pressure rise and fan efficiency.

| NO. | Bn | β/° | Δβ/° | t/mm | ϕ/° | $P_R$ | η |
|---|---|---|---|---|---|---|---|
| 1 | 17 | 35 | 1 | 1 | 57 | 348 | 57.7% |
| 2 | 17 | 40 | 2 | 1.5 | 60 | 386 | 60.4% |
| 3 | 17 | 45 | 3 | 2 | 63 | 410 | 60.7% |
| 4 | 17 | 50 | 4 | 1~2 | 68 | 707 | 69.1% |
| 5 | 18 | 35 | 2 | 1~2 | 63 | 211 | 46.5% |
| 6 | 18 | 40 | 1 | 2 | 68 | 273 | 48.6% |
| 7 | 18 | 45 | 4 | 1.5 | 57 | 511 | 64.0% |
| 8 | 18 | 50 | 3 | 1 | 60 | 733 | 71.5% |
| 9 | 19 | 35 | 3 | 1.5 | 68 | 195 | 44.9% |
| 10 | 19 | 40 | 4 | 1 | 63 | 486 | 64.4% |
| 11 | 19 | 45 | 1 | 1~2 | 60 | 476 | 61.6% |
| 12 | 19 | 50 | 2 | 2 | 57 | 474 | 61.6% |
| 13 | 20 | 35 | 4 | 2 | 60 | 210 | 53.6% |
| 14 | 20 | 40 | 3 | 1~2 | 57 | 314 | 54.7% |
| 15 | 20 | 45 | 2 | 1 | 68 | 607 | 67.0% |
| 16 | 20 | 50 | 1 | 1.5 | 63 | 574 | 64.5% |

In the orthogonal method, $k_i$ and $R$ are used to evaluate the influence of specific factors on indicators at their level, defined as follows:

$$k_i = \sum_1^n \eta_i \tag{6}$$

$$R = max(k_1, k_2, k_3, k_4) - min(k_1, k_2, k_3, k_4) \tag{7}$$

Table 7 can be obtained from Equations (6) and (7). According to the R-value in Table 8, the primary and secondary order of the factors affecting the static pressure rise in the fan is β > t > Δβ > Bn > ϕ. The impeller outlet installation angle has the greatest influence, and the diffuser tilt angle has the smallest influence. Based on the $k_i$ values, the optimal combination of factors is Bn = 17, β = 50°, Δβ = 4°, t = 1 mm, and ϕ = 60°.

**Table 7.** Range analysis of MIX-130 fan pressure rise.

| No. | Bn | β/° | Δβ/° | t/mm | ϕ/° |
|---|---|---|---|---|---|
| $K_1$ | 1851 | 964 | 1671 | 2174 | 1647 |
| $K_2$ | 1728 | 1459 | 1678 | 1666 | 1805 |
| $K_3$ | 1631 | 2004 | 1652 | 1367 | 1681 |
| $K_4$ | 1705 | 2488 | 1914 | 1708 | 1782 |
| R | 220 | 1524 | 262 | 807 | 158 |

**Table 8.** Range analysis of the MIX-130 fan Efficiency.

| No. | Bn | β/° | Δβ/° | t/mm | ϕ/° |
|---|---|---|---|---|---|
| $K_1$ | 2.478 | 2.028 | 2.324 | 2.606 | 2.380 |
| $K_2$ | 2.366 | 2.281 | 2.355 | 2.534 | 2.471 |
| $K_3$ | 2.248 | 2.533 | 2.317 | 2.245 | 2.361 |
| $K_4$ | 2.282 | 2.666 | 2.512 | 2.318 | 2.296 |
| R | 0.231 | 0.638 | 0.194 | 0.361 | 0.175 |

Table 8 can also be obtained from Equations (6) and (7). According to the R value in Table 7, the primary and secondary order of factors affecting fan efficiency is β > t > Bn > Δβ > ϕ. The impeller outlet installation angle has the greatest influence, and the diffuser tilt angle has the smallest influence. Based on the $k_i$ values, the optimal combination of factors is Bn = 17, β = 50°, Δβ = 4°, t = 1 mm, and ϕ = 60°.

The combination of factors and levels that give the fan the highest efficiency and hydrostatic boost is the number of blades Bn = 17, outlet installation angle β = 50°, outlet installation angle increment Δβ = 4°, blade thickness t = 1 mm, and diffuser tilt angle φ = 60°.

Based on the optimization results, the MIX-130 fan is redesigned with the optimal parameters, and the performance of the optimized fan is analyzed using the numerical method (Section 2).

As depicted in Figure 5a, the static pressure rise and efficiency of the original and optimized fans change with the flow rate. The static pressure rise and efficiency of the optimized fan are higher than that of the original fan.

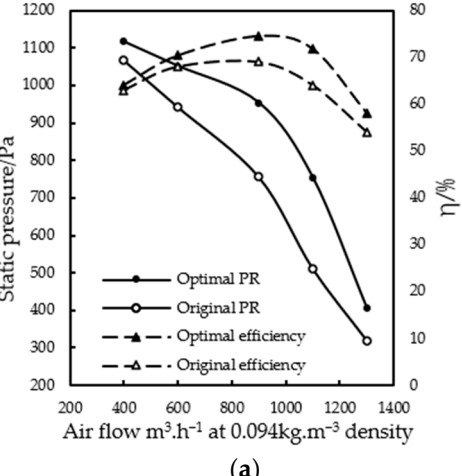 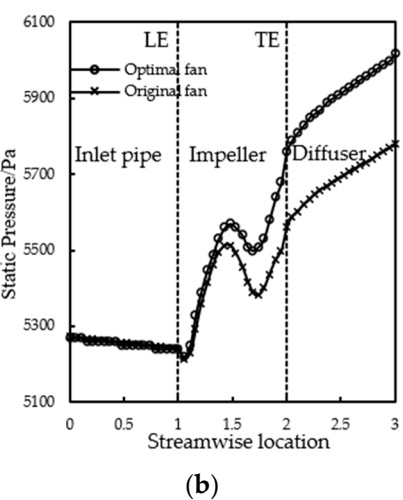

(**a**)                                                                                       (**b**)

**Figure 5.** (**a**) Comparison of static pressure rise and fan efficiency before and after optimization; (**b**) Fan static pressure distribution along the streamwise.

Under the design flow rate, the static pressure rise in the original fan is 511 Pa, and the static pressure rise in the optimized fan is 754 Pa, which increases by 47.5%. The efficiency of the original impeller is 64%, and that of the optimized fan is 72%, an increase in 8%.

Figure 5b compares the streamwise static-pressure distribution between the optimized and original fans. After the inlet pipe, the static pressure in the optimized fan is higher than that in the original fan. In the impeller section, the static pressure of the optimized fan is increased to 10%, and that of the original fan is increased to 6%. In the diffuser section, the static pressure boost of the optimized fan is 4.5%, and that of the original fan is 3.9%. These findings reveal that the optimization of the impeller and diffuser is highly effective.

Figure 6 illustrates the static pressure distribution of the impeller of the original and optimized fans at a flow rate of 0.8–1.2 Q. The static pressure on the impeller increases continuously from the leading edge to the trailing edge of the blade, indicating that the booster effect of the impeller is ideal. The static pressure of the optimized impeller is higher than that of the original impeller under the same flow rate. After optimization, the static pressure distribution on the suction surface of the impeller is more uniform, the pressure gradient decreases, and the airflow stability increases, so flow separation does not readily occur.

Figure 7 illustrates the aerodynamic distribution on the blade. The force distribution uniformity on the optimized impeller increases, and maximum aerodynamic force is exerted on the suction surface of the blade. There are relatively concentrated aerodynamic forces at the front edge of the pressure and the trailing edge of the suction surface. However, the value is small, consistent with expectations. However, the aerodynamic distribution of the blade before the optimization is irregular, indicating that the airflow near the blade is relatively chaotic.

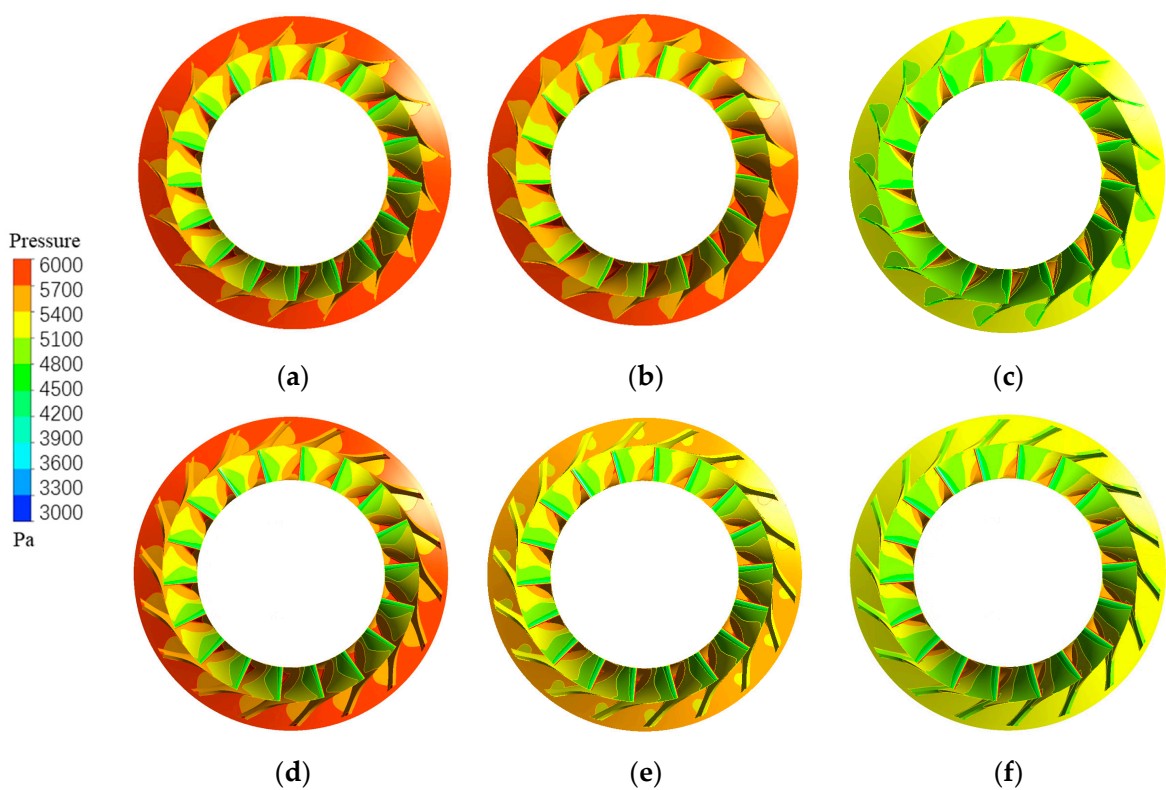

**Figure 6.** Static pressure distributions in impellers of original and optimized fans: (**a**) optimize impeller 0.8 Q; (**b**) optimize impeller Q; (**c**) optimize impeller 1.2 Q; (**d**) original impeller 0.8 Q; (**e**) original impeller Q; (**f**) original impeller 1.2 Q.

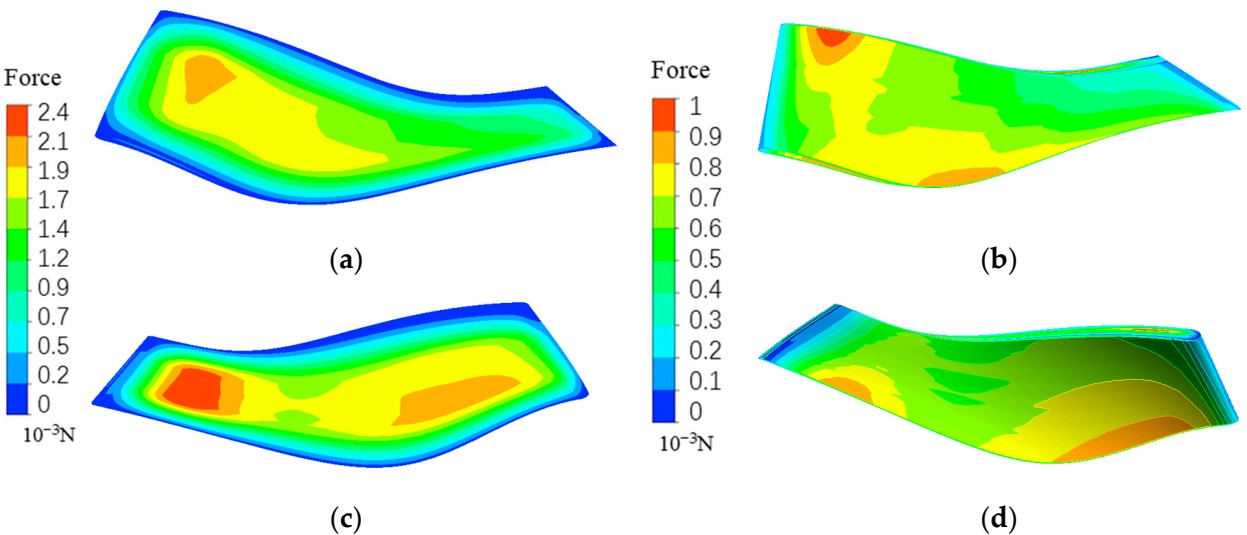

**Figure 7.** Blade force distribution (**a**) optimize blade pressure surface; (**b**) original blade pressure surface; (**c**) optimize blade suction surface; (**d**) original blade suction surface.

Figure 8 illustrates the static pressure distribution in the middle surface of the diffuser at a flow rate of 0.8–1.2 Q for the original and optimized fans. The static pressure in the fan increases with the increase in the diffuser diameter, exhibiting a good distribution, indicating that the diffuser has a high static-pressure transition ability. Under the same flow, the static pressure of the optimized diffuser is greater than that of the original diffuser, revealing that the optimization of the diffuser inclination angle is highly effective.

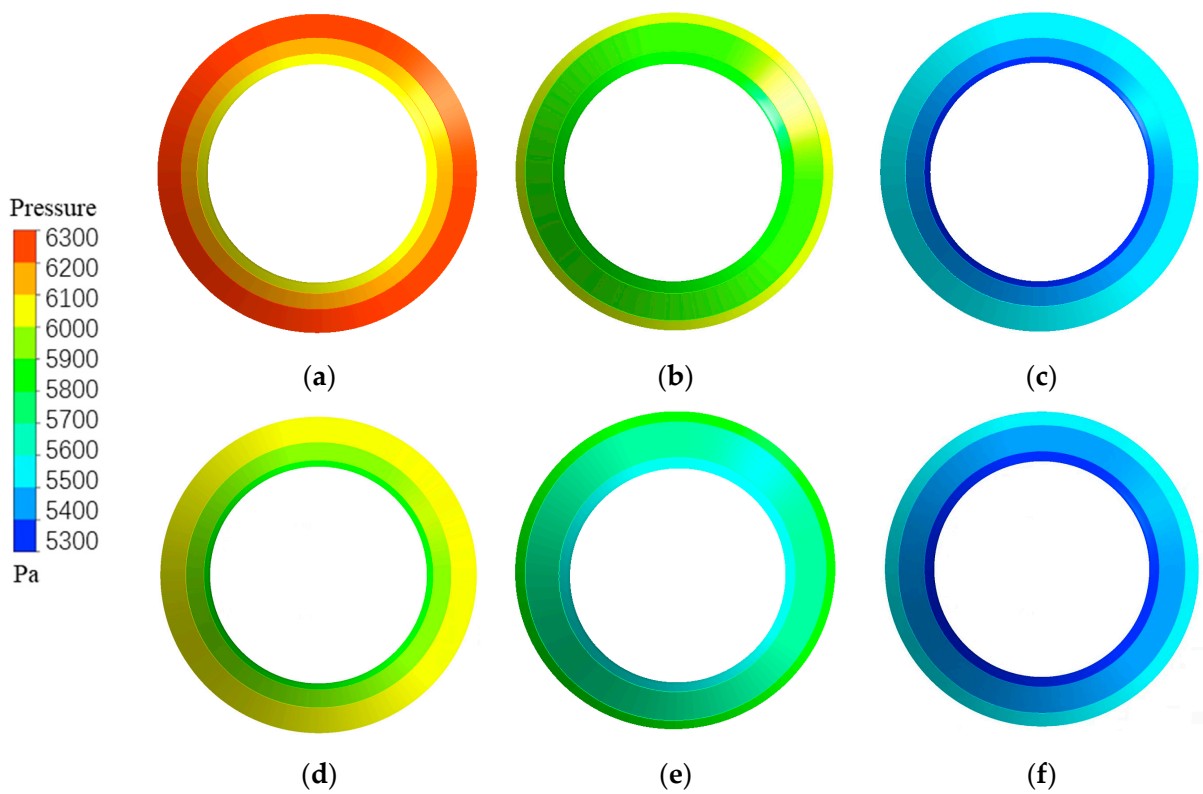

**Figure 8.** Static pressure distributions in the diffuser of original and optimized fans: (**a**) optimize diffuser 0.8 Q; (**b**) optimize diffuser Q; (**c**) optimize diffuser 1.2 Q; (**d**) original diffuser 0.8 Q; (**e**) original diffuser Q; (**f**) original diffuser 1.2 Q.

## 4. Fan Test

Figure 9 illustrates the optimized MIX-130 fan, including the overall structure of the fan and the structure of each component.

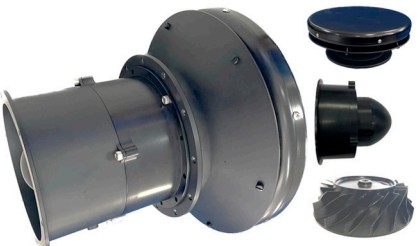

**Figure 9.** Optimized MIX-130 fan.

### 4.1. Test Method and Test Bench

Several mature methods and devices are available for testing fan performance at normal temperatures and pressure. However, measuring fan performance in a stratospheric, low-temperature, low-pressure environment requires a large enough environmental test chamber and specially customized test equipment. The cost is high, and the testing conditions are not currently available. This study proposes a method for testing and verifying fan performance at a high altitude. The fan's performance curve at a high altitude is obtained by measuring it at normal temperature and pressure, combined with the law of similarity conversion and numerical analysis. The accuracy of the performance curve obtained by the conversion is verified by measuring the maximum static pressure value of the fan at a high altitude.

Because the design point of the MIX-130 fan is the stratospheric, low-temperature, low-pressure condition, the full-speed test condition is not available when testing under

the ground's normal temperature and pressure conditions, so the speed should be reduced for testing to ensure that the fan power is consistent with the high-altitude condition. The similarity principle converts fan performance under different working conditions [40,41]. The similarity principle is defined by Equations (8)–(10):

$$Q = Q_d n/n_d = NQ_d \tag{8}$$

$$p = p_d(n/n_d)^2(\rho/\rho_d) = N^2 p_d\left(\frac{\rho}{\rho_d}\right) \tag{9}$$

$$P = P_d(n/n_d)^3(\rho/\rho_d) = N^3 P_d\left(\frac{\rho}{\rho_d}\right) \tag{10}$$

where $Q_d$, $p_d$, $P_d$, $n_d$ and $\rho_d$ are the flow rate, pressure, power, speed, and medium density of the fan at the design point, and $Q$, $p$, $P$, n, and $\rho$ are the operation values of the fan at other working conditions.

Based on Equation (10), the rated power $P_d$ of the fan can be reached when the fan speed reaches 40% of the maximum speed under normal temperature and pressure on the ground.

The performance test of the MIX-130 fan at room temperature and pressure was conducted on the fan test bench by the AMCA (Air Movement and Control Association) specification, as depicted in Figure 10a, which was used to measure and record the fan performance curve. The test was conducted under an ambient temperature of 298 K, pressure of 0.1 atm, and relative humidity of 50%. The fan speed was set at 40% maximum rotation.

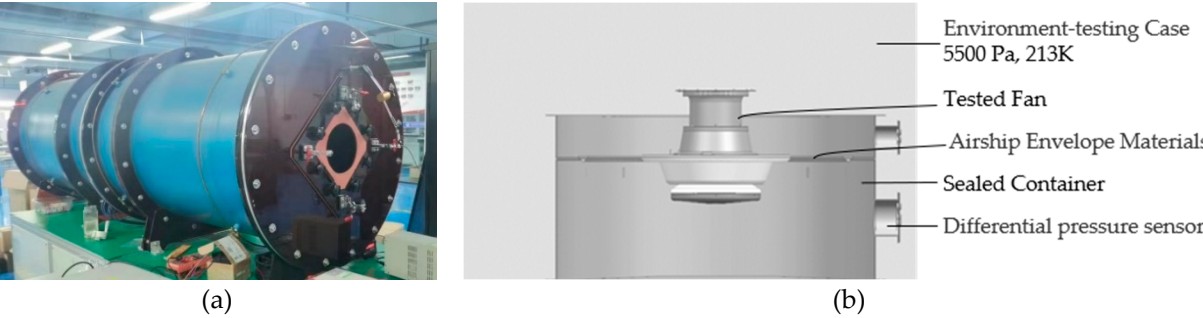

(a)            (b)

**Figure 10.** (**a**) Fan performance test bench for the ground condition; (**b**) Fan performance test bench for stratospheric condition.

The performance test of the MIX-130 fan at low temperature and low pressure is conducted on the test bench, as depicted in Figure 10b. Through this test bench [42], the maximum static pressure of the fan can be measured under any working condition at a high altitude. The test was conducted under a temperature of 213 K and pressure of 5500 Pa. The fan speed was set to the maximum speed.

### 4.2. Comparison of Test and Simulation Results

The curve of static pressure variation with the flow of the MIX-130 fan at normal temperature and pressure (P-Q curve for short) can be obtained using the numerical analysis model and method in Section 2 and changing the boundary conditions into the ground test environment. As depicted in Figure 11a, the two P-Q curves represent the numerical simulation and test results. The MIX-130 fan has a maximum static pressure of 2550 Pa and a maximum flow rate of 600 m$^3$/h at 40% of the maximum speed on the ground, and there is no saddle-shaped area in the entire P-Q curve. Based on the analysis, the average error between the numerical simulation and the experimental test is 3%, confirming the accuracy of the numerical analysis.

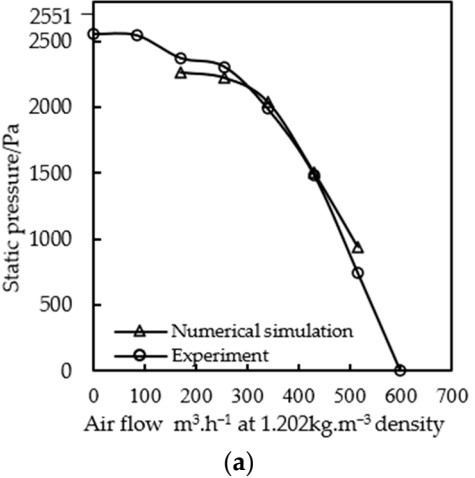
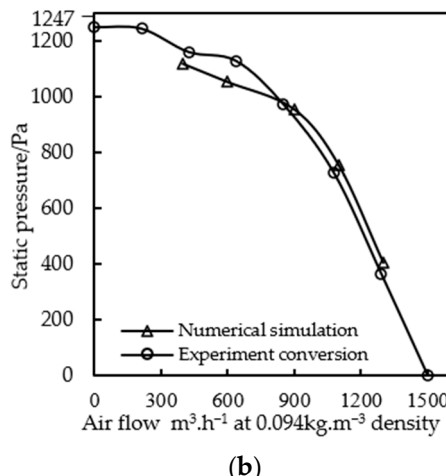

**Figure 11.** (**a**) The ground performance curve of the MIX-130 fan; (**b**) Stratospheric performance curve of the MIX-130 fan.

Based on Equations (8) and (9), the measured P-Q curve on the ground of the MIX-130 fan can be converted into the P-Q curve of stratospheric working conditions. As depicted in Figure 11b, the two curves are the stratospheric P-Q curves obtained from the numerical analysis in Section 2, and the stratospheric P-Q curves are measured and converted. Based on the analysis, the average error between the two is 4.4%, confirming that the similarity law is more accurate for converting fan performance in different working conditions.

The maximum static pressure of the MIX-130 and MAX45016 fans under stratospheric conditions was tested when the environment was stable at a temperature of 213 K and a pressure of 5500 Pa using the test bench in Figure 10b. As depicted in Figure 12, the test was completed in the Environmental Laboratory of Aerospace Information Research Institute, Chinese Academy of Sciences. The maximum static pressure of the MAX45016 and MIX-130 fans are 503 and 1210 Pa.

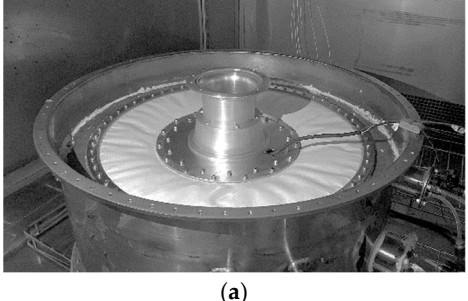
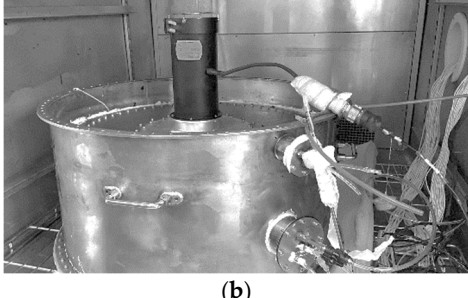

**Figure 12.** (**a**) MIX-130 fan test status; (**b**) MAX45016 fan test status.

Figure 13 illustrates the P-Q curve of the MAX45016 and MIX-130 fans under stratospheric conditions. The P-Q curve of the stratospheric working condition of the MAX45016 fan can be converted from the P-Q curve at normal temperature and pressure using Equations (8) and (9), with a maximum static pressure of 506 Pa. The stratospheric P-Q curve of the MIX-130 fan is obtained by numerical simulation in Section 2, and its maximum static pressure is 1245 Pa. The error between the maximum static pressure obtained from the test and the data obtained from the numerical analysis is small, confirming the accuracy of the numerical method.

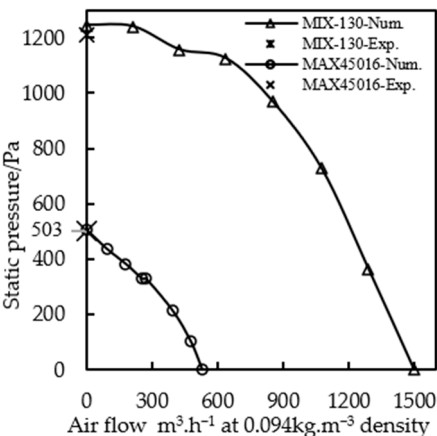

**Figure 13.** Performance comparison of fans in stratospheric conditions.

In stratospheric conditions, the maximum flow rate of the MIX-130 fan increases by 184%, and the maximum static pressure increases by 146% compared with the existing MAX45016 aviation fan. The performance of the MIX-130 fan can satisfy the requirements of a stratospheric airship in the first stage.

## 5. Conclusions

In this study, the $L_{16}(4^5)$ orthogonal test with five factors and four levels was used to optimize the performance of the mixed-flow fan used for a stratospheric airship, and a high optimization effect was obtained. The performance curves of the fan underground and stratospheric conditions were obtained using two different fan performance test stands, and the accuracy of the numerical calculation was verified by comparison.

The conclusions of this manuscript are as follows.

(1) The optimal fan design based on an orthogonal test has a significant effect. The results demonstrate that the static pressure lift of the optimized fan increases by 47.5%, and the efficiency increases by 8%.
(2) The average error between the performance curves obtained by testing the MIX-130 fan underground conditions and the numerical calculations is 3%, which proves the high accuracy of the CFD method and test means of this manuscript in obtaining the performance curves of the mixed-flow fan.
(3) The ground fan's performance curve is converted to the stratospheric performance curve by the law of similarity. Numerical and experimental methods confirm the accuracy of this conversion. This method can provide a reference for determining the performance curves of the fan at different altitudes.
(4) The MIX-130 fan can produce a flow rate of 1100 m$^3$.h$^{-1}$ and a static pressure lift of 754 Pa at the design point, which can meet the current requirements of stratospheric airships. The fan structure design with a mixed-flow impeller and blade-less diffuser is highly effective. The fan structure optimization and testing methods presented in this manuscript can provide a reference for designing and testing stratospheric airship fans.

For future works, In terms of fan optimization, we will carry out fan parameter optimization based on an algorithm [22,28,43] to continuously optimize fan performance; Practical application of fan: It is expected that in the summer of 2023, the MIX-130 fan will be mounted on stratospheric airships, overpressure balloons and other aerostats to carry out several studies in stratospheric environment, including the study on fan performance in low temperature and low-pressure environment, and the study on the sensitivity of fan parameters to the height regulation of overpressure balloons.

**Author Contributions:** Conceptualization, W.Q. and T.Z.; methodology, W.Q.; software, W.Q.; validation, W.G. and C.C.; investigation, Z.H.; data curation, W.Q.; writing—original draft preparation, W.Q.; writing—review and editing, T.Z. and Z.H.; visualization, C.C. All authors have read and agreed to the published version of the manuscript.

**Funding:** This research was funded by "CAS strategic leading science and technology project (Class A), XDA17000000"; CAS science and disruptive technology research pilot fund, E1Z220010F.

**Data Availability Statement:** Not applicable.

**Acknowledgments:** We thank Jiangsu Jiu Gao Tech Co., Ltd. for providing testing support for this study.

**Conflicts of Interest:** The authors declare no conflict of interest.

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
