# Peer review of "Optimization Design and Experimental Verification for the Mixed-Flow Fan of a Stratospheric Airship"

_aerospace, doi:10.3390/aerospace10020107_

Round 1

Reviewer 1 Report

Please read the attachment. Thank you. 

Reviewer 2 Report

The paper describes an interesting study concerning the optimization of turbines operating in stratospheric conditions.

A few notes on the document:

1) It is regrettable that the authors seem to focus only on Chinese references and are not open to developments made around the world. An enrichment of the bibliographical references by non-Chinese references, if only by consulting the journal Aerospace, could moderate this negative impression.

2) The optimization in paragraph 3.1 is based on conjectures and an empirical process without theoretical foundation which could guide future research on this topic. It is important to specify the choice of this point cloud and the limit values set for each parameter. Without this, the developments will appear as a study at a given operating point of a given system without the possibility of transposing these results to other scenarios or other operating points, whereas the authors claim in points (3) and (4) from the conclusion that: (I quote)"This method can provide a reference for determining the performance curves of the fan at different altitudes.

The stratospheric performance curve of the MIX-130 fan can provide a basis for pressure control of a stratospheric airship."

3) The intermediate equations in the systems of equations (1-3) must be modified by writing for example on the right (h=11000m) and putting the value of the corresponding parameter on the left.

Round 2

Reviewer 1 Report

Dear Editor and Authors:

Thank you for providing the point-to-point response.

The authors have corrected and answered all comments and questions. The manuscript sounds perfect now. The reviewer strongly suggest it be accepted for publication. 

Thank you for reading. 

Sincerely yours,

The Reviewer.